# Abnormal Gait Detection Using Wearable Hall-Effect Sensors

**DOI:** 10.3390/s21041206

**Published:** 2021-02-09

**Authors:** Courtney Chheng, Denise Wilson

**Affiliations:** 1Department of Electrical Engineering, University of Washington Bothell, Bothell, WA 98011, USA; 2Department of Electrical and Computer Engineering, University of Washington Seattle, Seattle, WA 98195, USA; denisew@uw.edu

**Keywords:** gait monitoring, Hall-effect sensors, magnetic sensors, wearable sensors, gait irregularities, stride, cadence, stride width

## Abstract

Abnormalities and irregularities in walking (gait) are predictors and indicators of both disease and injury. Gait has traditionally been monitored and analyzed in clinical settings using complex video (camera-based) systems, pressure mats, or a combination thereof. Wearable gait sensors offer the opportunity to collect data in natural settings and to complement data collected in clinical settings, thereby offering the potential to improve quality of care and diagnosis for those whose gait varies from healthy patterns of movement. This paper presents a gait monitoring system designed to be worn on the inner knee or upper thigh. It consists of low-power Hall-effect sensors positioned on one leg and a compact magnet positioned on the opposite leg. Wireless data collected from the sensor system were used to analyze stride width, stride width variability, cadence, and cadence variability for four different individuals engaged in normal gait, two types of abnormal gait, and two types of irregular gait. Using leg gap variability as a proxy for stride width variability, 81% of abnormal or irregular strides were accurately identified as different from normal stride. Cadence was surprisingly 100% accurate in identifying strides which strayed from normal, but variability in cadence provided no useful information. This highly sensitive, non-contact Hall-effect sensing method for gait monitoring offers the possibility for detecting visually imperceptible gait variability in natural settings. These nuanced changes in gait are valuable for predicting early stages of disease and also for indicating progress in recovering from injury.

## 1. Introduction

How individuals walk (i.e., gait) can vary as a result of walking surface, individual body type, injury, disease, fatigue, and a variety of other factors. Abnormalities or irregularities in gait can both predict and indicate health problems. For instance, disturbances in gait have been found to be early predictors of fall risk in the elderly and of physical disability [1,2]. Further, irregularities in the way individuals walk have proven to be accurate diagnostic indicators of a range of neurological diseases, including Parkinson’s disease, Huntington’s disease, and Alzheimer’s disease as well as of impaired mobility in older adults [1,3,4,5]. Gait also changes as individuals recover from injury or adapt to disability, and monitoring gait during these processes provides valuable information regarding the success or failure of rehabilitation strategies. For example, among children with cerebral palsy, gait is monitored to guide and improve the viability of adapted walking strategies during rehabilitation training in children [6]. Thus, monitoring the simple act of walking can provide important information for detecting, treating, and healing from injury, for early detection and diagnosis of disease, and for identifying risk of injury among the elderly and other vulnerable populations.

### 1.1. Gait Parameters

A wide range of parameters are used to characterize gait, but stride width and stride width variability seem especially important for the diagnosis and monitoring of injury and disease. A stride is simply a sequence of two steps. Stride width is the distance between two imaginary parallel lines, passing through the heel of the foot [7] during a stride. Variability in stride width is typically described in terms of the coefficient of variation (i.e., standard deviation divided by the mean) or standard deviation in stride width for a group of similar strides [1]. Such variability has been found to be the most reliable gait variable in assessing gait variability in patients with Parkinson’s disease [8]. Increased step width variability is also often associated with diminished balance control and gait stability [1]. Extreme step width variability (either large or small) contributes to fall risk [9] among older adults but is dependent on gait speed and cadence. For those who walk slowly, there is no increase in fall risk, but for those who walk at normal speeds, risk increases as stride width variability becomes extreme [3,9]. This dependence on cadence (steps per minute) or gait speed underscores the role that assessing cadence can play in overall gait monitoring. Studies have shown that while gait speed and step length decrease with age and stride width increases, cadence remains relatively constant, ranging on average between 115 and 120 steps per minute for normal gait [10]. Similarly, studies of individuals with Parkinson’s disease have demonstrated no significant differences in cadence from typically developed (healthy) adults [11]. Thus, in many scenarios, cadence serves as a context for studying other gait parameters. Knowing cadence allows variations in other gait parameters to be assessed more accurately. This is not to say, however, that cadence in and of itself has no value. In fact, among certain physical and psychological conditions, cadence does indeed change and serve as one of multiple indicators that there may be a problem. For instance, depressed persons often walk more slowly [12]. Among physical impairments, loss of vision has been associated with decreased walking speed, among other factors [3]. Cadence can also support the prevention of injury. For example, among runners at risk for tibial stress fractures, decreasing cadence reduces risk of such fractures [13]. Thus, cadence is valuable on its own for detecting gait disturbance and also when used as a context for assessing changes in other gait parameters such as stride width.

### 1.2. Gait Monitoring and Analysis

Because the normalness of gait is affected by a wide range of factors, not all of which involve disease or injury, a vast majority of gait irregularity studies have been restricted to clinical settings. In clinical settings, extraneous factors can be controlled in order to focus on detecting and assessing gait disturbances that stem from physical injury, psychological origins, or central nervous system disease. In the clinical setting, gait analysis is typically conducted with camera monitoring systems that visually capture gait and are often combined with force platforms which capture ground reaction forces on the feet as an individual walks. These multi-sensor, three-dimensional systems capture an immense amount of data that are typically evaluated off-site by a qualified physician [14]. Clinical gait monitoring equipment is often only found in gait or specialized locomotion laboratories and requires significant capital investment and lengthy setup and post-processing times [15]. On the other end of the spectrum, wearable gait monitoring systems used outside of a clinical setting are much less expensive and require far less power. However, these systems are inherently vulnerable to interference from a wide range of environmental and compliance factors. Further, the sensors on these devices often offer less accuracy than those in clinical settings and the end result is that they generate data that are less successful at identifying abnormalities and irregularities in gait.

Nevertheless, wearable sensor systems support a more practical health care paradigm that expands and equalizes access to quality health care across different populations. Systems that are capable of delivering health care services at any time and location have significant global implications for supporting personalized health care, disease prevention, point-of-care diagnosis, and treatment of chronic disease conditions [16,17]. Wearable sensor systems also open new possibilities for continuous monitoring, timely decision making, and reductions in unnecessary hospitalization while also being non-invasive or minimally invasive to their subjects [16,18]. Moreover, wearable sensors have the unique and valuable advantage of monitoring subjects in their natural setting. Especially relevant to gait monitoring, wearable sensors can drive down the health care costs of an aging population, both by providing better preventative care and greater diagnostic capability [17,18]. Wearable gait monitoring sensors and systems are also capable of providing quantitative and repeatable results over extended time periods [19]. As a result, gait analysis using wearable sensors has drawn researchers from many fields interested in detection of human locomotion.

Existing wearables have demonstrated their value to health care through their use as a safe, cost-effective health care device in tracking physical activity [20]. There are several popular existing health-monitoring wearables on the market including the Fitbit, Apple Watch, and the Samsung Galaxy Active watch. A systematic review of these personal health care monitoring devices concluded that regardless of age, sex, and health status, people using wearables improved their amount of physical activity and daily steps [20]. Wearable devices can also be used to develop a personalized intervention based on the user’s goals and activity. Generally, users agree that wearable devices have positive value in promoting a healthy lifestyle [20] and the data collection function in these devices supports the development of customized interventions to promote wellness. In addition to promoting wellness, wearables have also provided early detection of anomalies and disease and facilitated early treatment and prevention of more serious health problems. A recent study found Fitbits and other wearables which monitor sleep and resting heart rate to be effective in detecting and tracking influenza outbreaks [21]. Ample anecdotal testimony attests to the ability of wearables to detect extremes in heart rate that prompt wearers to seek immediate medical treatment and avoid serious if not fatal health consequences of these episodes [22]. When used in conjunction with medical treatment, wearables can fast track adjustments and improvements in treatment. For example, wearable monitors have been successfully paired with medical treatment to monitor and optimize clinical interventions on patient mobility [23]. Despite problems with compliance and imperfect data collection, wearables such as the Fitbit have also been successfully used to help depressed, alcohol-dependent women engage in lifestyle physical activity to resist their urge to drink during recovery [24]. Thus, it is no surprise that personal health care wearables and their potential to support improved and expanded access to health care have contributed substantially to the 14% increase in over 86 million wearable devices sold globally in the second quarter of 2020 [25].

### 1.3. Gait Monitoring Technologies

Gait monitoring technologies in clinical settings are typically not worn on the body and are dominated by camera-based systems and force-sensitive platforms. Cameras capture human movement and then utilize video and image processing to extract gait parameters [26]. These systems have the advantage of accurately capturing many nuances of observable gait but are expensive and processing of image data is often hindered by variations in the spectrum and intensity of ambient illumination. Further, camera-based methods, especially those that capture gait in three-dimensions, generate large amounts of data which limit the opportunity to process the data in real time and create bandwidth and storage issues at the back end of these systems. As importantly, while camera-based systems provide prolific and accurate information about disturbances in gait, they are inherently limited in identifying the source of those disturbances. For example, an individual who is limping may be doing so to avoid pain for any number of reasons which remain largely opaque and inaccessible to camera-based systems. To address this lack of contact-based information, force-sensitive platforms are a common supplement or alternative to camera-based systems in clinical settings. The force-sensitive platform contains force sensors embedded in walking pads or platforms. Each pressure pad contains large arrays of pressure sensors placed in a grid array which are then interconnected with each other [26]. These platforms track in considerable detail the forces exerted by each foot as an individual walks and can provide not only accurate measures of step length, stride length, stride width, cadence, and similar parameters, but also monitor abnormalities in foot pressure that can be indicative of unhealthy posture, injury, or disease. While these high-resolution data are invaluable for providing greater insight into the forces on the feet and subsequent gait disturbances, they only monitor the feet when they are in contact with the floor (i.e., platform) and therefore lack information generated during the swing phase of gait [26,27].

A number of wearable sensors for gait monitoring have been developed over the past two decades. A representative subset of these wearables is summarized in Table 1. A more thorough review of wearable sensors for gait monitoring can be found in [15,27,28].

A vast majority of these wearables make use of a gyroscope (for measuring angular velocity and orientation), accelerometer, or inertial measurement unit (IMU—which combines a multi-axis gyroscope and accelerometer and may also include a magnetometer) and are worn on the foot (shoe). For example, the GaitShoe uses an extensive suite of sensors including three accelerometers, three gyroscopes, four force sensors, and electric field sensors integrated into a sandal-like shoe that enables gait data to be collected unobtrusively over long periods of time and with minimal interference to normal gait [19]. Data regarding orientation, acceleration, and velocity collected from these accelerometers, gyroscopes, and IMU-based wearables on the foot have been successfully used to determine specific gait parameters including center of pressure in the foot [46], step length, and stride width [38]. Gyroscopes and accelerometers have been successfully employed to differentiate multiple different phases of human gait [41,46]. Wearables on the foot have also been used to fully capture and analyze foot motion [29,42].

Accelerometers and gyroscopes are not the only sensors integrated into these wearable sensors. For example, researchers have developed a wearable gait tracking device for gait characteristics by implementing a set of force-sensitive resistors (FSR) to detect foot pressure at various locations on the underside of the foot and use that information to categorize six different stride positions during human gait known as heel strike, foot flat, heel off, terminal stance, and toe off [54]. Ultrasonic range sensors [38,52], infrared sensors [42], electric field [19], and electromyography (EMG) sensors for the detection of muscle activity during gait [33] have also been demonstrated in wearable gait sensors. These myriad sensors have been applied to the measurement of both temporal and spatial gait parameters achieving accuracies on the order of cm and sub-cm for spatial parameters (e.g., stride length, stride width, toe and heel position) and millisecond accuracy for temporal parameters (e.g., gait speed, cadence). Furthermore, a number of wearable sensors have also been demonstrated specifically for differentiating abnormal gait from normal gait (Table 2) with applications ranging from monitoring patients with Parkinson’s disease [19] to recovery and rehabilitation from knee surgery [37].

While the research and development space for wearable gait monitoring sensors may seem crowded, human gait is complex and multi-faceted, which leaves ample opportunity for other wearables to address gaps that remain in the literature. This study seeks to fill the “Fitbit” gap for wearable gait monitors by targeting an unobtrusive, low-power, and durable wearable based on a Hall-effect sensor that can act not only as a monitor of healthy gait but also as an early indicator of gait disturbances, irregularities, or abnormalities. To reach these goals, the wearable presented herein focuses specifically on measuring cadence and stride width variability which can serve as an important monitor of Parkinson’s disease progression [5] and recovery from injury [3] and surgery [55] as well as an early indicator of Alzheimer’s disease [56,57] and increased risk of falling [9,58,59]. While other wearables have been developed to measure these parameters (Table 3), our approach focuses on reducing the footprint of operation while simultaneously increasing the precision of measurement using the Hall-effect approach.

Hall-effect sensors have been demonstrated in past research efforts for wearable systems. For example, four magnets embedded in silicon rubber were mounted in shoes above four Hall-effect sensors to measure the loading on the feet in normal and shear directions, which is often critical for properly treating diabetic foot ulcers [60]. Hall-effect sensors have also been used to estimate knee angle in prosthetics. A low-power magnetic measurement system based on two Hall-effect sensors and a magnet has been integrated into a smart knee prosthesis to accurately measure knee flexion-extension. Researchers used linear and locally linear neuro-fuzzy estimators to translate the magnetic measures into knee flexion-extension while reducing power consumption for doing so by almost three-fold [61].

### 1.4. This Study

Given these advantages of the Hall-effect approach, this study seeks to explore the suitability of the Hall-effect sensor for capturing cadence and leg gap as a proxy for stride width and stride width variability. The feasibility of using this system for detecting gait disturbances is evaluated using a prototype wearable sensor system that is mounted on the upper leg (e.g., thigh) just above the knee. Upper leg wear is less vulnerable to abuse and premature wear and tear compared to sensors that are mounted on the bottom or side of shoes or the feet due to fewer mechanical stresses experienced along the upper leg. This combined with the fact that Hall-effect sensors have no moving parts offers the possibility for a much longer lifetime in a corresponding wearable than classical alternatives that use accelerometers or gyros.

## 2. Materials and Methods

A block diagram of the wearable sensor system used to monitor gait at the knee and upper leg is shown in Figure 1. A continuous-time ratiometric linear analog Hall-effect sensor manufactured by Allegro Microsystems (A1302) [62] was paired with a 2.5 cm diameter (0.8 cm inner diameter) N52 Ni-Cu-Ni (Neodymium) ring magnet from KJ Magnetics. The sensor produced a voltage output with sensitivity ranging between 1.3 mV/G and 1.6 mV/G at a typical supply voltage of 5 V and maintained a baseline (zero magnetic field) output voltage equal to one half of the supply voltage. The output voltage increased or decreased from this baseline value based on the polarity and the intensity of the ambient magnetic field. Sensor precision was determined by calibrating the Hall-effect sensor by placing the face of the magnet parallel to the sensor and adjusting the distance between the two. At nominal leg gap widths of approximately 1.6 cm (corresponding to a distance between sensor and magnet of approximately 1 cm), the precision of the sensor was limited by the 10-bit analog to digital converter on the Arduino microcontroller, resulting in 0.07 mm precision. The range of the sensor was estimated as 5 cm and was identified as the point at which a 0.16 cm increment in the magnet–sensor distance produced a change in the digital sensor output of less than two states.

The sensor was mounted on the outside of a lightweight knee brace (Figure 2) such that the face of the sensor was parallel to the leg and in close proximity to the opposite leg during mid-stance when the feet are together. The magnet was mounted directly to the clothing on the opposite leg such that flat face of the ring magnet was parallel to the opposite leg and directly across from the Hall-effect sensor (Figure 2). One 9 V battery provided power to an Arduino Uno microcontroller board that provided a regulated 5 V output to the Hall-effect sensor, collected data from the sensor, timestamped the data, and transferred it to a 2.4 GHz Xbee for wireless transmission to a laptop computer. The sensor data were monitored at 575,000 baud, recorded, saved, and analyzed in Matlab R2020A using the Statistics and Machine Learning toolbox. The wearable prototype was used to evaluate whether this approach to gait monitoring could serve as a low-power, high-resolution means to capture a proxy for stride width and cadence. The footprint of the prototype system is quite large and would reduce by a factor of four or more in a fully integrated wearable system.

The goal of testing and data collection at this phase in the research was proof of concept, leading to two research questions:RQ1: Does leg gap width captured using the wearable system make a sufficient proxy for stride width in terms of differentiating abnormal and irregular strides from normal strides?RQ2: Can the wearable system capture cadence with sufficient accuracy to differentiate abnormal and irregular strides from normal strides?

To evaluate these two research questions, the gait monitoring system prototype was tested on four individuals walking with five different types of gait. Each individual walked six different segments consisting of approximately 10 m each on smooth concrete in an indoor environment, for a total of 60 m walked for each of the five types of gait per individual (Table 4). Abnormal strides were simulated by asking each individual to walk at a brisk or fast stride compared to their normal pace and then at a pace that was slow compared to what each individual considered normal or natural. Limping strides (on both left and right) strides were used to represent irregular strides and were simulated by asking each individual to walk with the left shoe off (right antalgic) and then with the right shoe off (left antalgic). Antalgic gait is a form of irregular gait which is typically developed to avoid pain by walking (i.e., a limp). It is defined as a gait where the stance (i.e., foot on the ground) phase of the affected leg is shortened relative to the swing phase as compared to normal or healthy gait. The subject spends most of their stance time with their weight being placed on the normal or healthy leg. In antalgic gait, the swing phase is increased on the affected side and may be shortened on the normal leg in order to get the normal leg back to the ground [33].

Among the five groups of strides and four subjects evaluated, the number of strides recorded for each segment varied with the type of gait, individual, speed of gait (cadence), and stride length. Raw data collected over the wireless Xbee connection were in the form of 10 bit digital signals representing sensor signals between 0 and 5 V. The ratiometric nature of the Hall-effect sensor forced the sensor output to *V*_dd_/2 (i.e., 511 or 512 as a 10 bit digital signal) at zero magnetic field where *V*_dd_ is the power supply (5 V) supplied to the sensor. The raw digital data were preprocessed to generate an analog voltage corresponding to the sensor output:*V*_out_ = (5/1023)∙Digital Output(1)

Once converted to a voltage, the leg gap width (as a proxy for stride width) was estimated using known properties of the magnet and of the Hall-effect sensor. The magnetic field is related to the distance *x* from the center of the magnet along an axis perpendicular to the flat disk surface (Figure 3) as follows:(2)$$B_{x}=\;\frac{B_{r}}{2}(\left(\;\frac{x+T}{\sqrt{R_{o}{}^{2}+{\left(x+T\right)}^{2}}}-\;\frac{x}{\sqrt{R_{o}{}^{2}+x^{2}}}\right)-\left(\;\frac{x+T}{\sqrt{R_{i}{}^{2}+{\left(x+T\right)}^{2}}}-\;\frac{x}{\sqrt{R_{i}{}^{2}+x^{2}}}\right)$$
where $R_{o}$ is the outer radius of the magnet (2.54 cm); $R_{i}$ is the inner radius of the magnet (0.794 cm); *x* is the distance from the sensor to the magnet surface, and *T* is the thickness of the magnet. $B_{x}$ is the magnetic field at *x*, and $B_{r}$ is the surface magnetic field of the magnet as specified by the manufacturer. Magnetic fields are in units of Gauss (G).

Once the magnetic field *B_x_* is known, the output of the Hall-effect sensor can be estimated as:*V_out_* = *S∙B_x_* + *V_dd_/2*(3)
where *S* is the sensitivity of the Hall-effect sensor. The distance *x* can be calculated numerically based on the sensor output using these relationships.

Finally, the leg gap width (i.e., the distance between the legs at the point where the sensors are mounted) is then estimated as:(4)$$Leg\;gap\;Width\;=\;x\;+\;T\;+\;W_{Sensor+Board}$$
where $W_{Sensor+Board}$ is the sum of the width of the sensor and the width of the printed circuit board on which the sensor is mounted.

In order to eliminate the impact of variations in swing between the two legs, leg gap width (i.e., leg gap) was calculated at the peak sensor output associated with each step involving only the swing of the left leg (i.e., mid-stance peak). In order to determine these mid-stance peaks, noise in the raw sensor data was first reduced by identifying local maxima in the sensor output that were at least 0.8 s apart. This minimum stride time corresponds to a maximum detectable cadence of 150 steps/min, well above the 115–120 steps/min at which a typical adult walks. Once these local maxima were identified, the sensor output was filtered by replacing all data that did not correspond to one of these local maxima with the value of the sensor baseline, essentially zeroing the non-maxima values out. Then, the local maxima were filtered by their prominence in order to eliminate spurious, low-signal maxima that did not represent mid-stance peaks. While this approach eliminated some valid mid-stance peaks, it ensured that very few false peaks remained in the data. The timestamps of the remaining peaks in the preprocessed data were then retained to calculate cadence and the raw sensor output voltage corresponding to the mid-stance peak locations retained to calculate leg gap width using Equations (1)–(4).

Strides per minute were calculated simply as the reciprocal of adjacent mid-stance peaks (in sec) and scaled to minutes:*Strides per minute = 60/(t_n_-t_n−_*_1_*)*(5)
where *t*_n_ is the time at which a mid-stance peak occurs for stride *n* while *t**_n−_*_1_ is the time at which a mid-stance peak occurs for the stride immediately before stride *n*. Since a stride consists of two steps, the cadence was then calculated as twice the strides per minute:*Cadence = 2∙Strides per minute*(6)

Once calculated and checked for normality, cadence and stride width were used to evaluate differences among stride types among all four individuals tested. The goal of data analysis was to understand whether the data collected by the wearable sensor system was capable of distinguishing abnormal gait (fast, slow) and irregular gait (right antalgic, left antalgic) from normal gait via differences in cadence, differences in stride width, differences in stride width variability, differences in cadence variability, or some combination thereof.

Appropriate statistical tests were then used to assess whether the null hypothesis that an abnormal or irregular gait for a given subject was the same as a normal gait could be rejected. For cadence, a Kruskal–Wallis test was used to assess whether cadence associated with each abnormal or irregular stride was significantly different from that associated with normal stride. The Kruskal–Wallis test is a non-parametric test that compares the medians rather than the means of different sample populations and does not rely on assumptions of normality for accuracy [63]. This same test was used to evaluate whether leg gap width for abnormal or irregular stride was significantly different than the leg gap width associated with normal stride for the same individual. The variability in leg gap width was evaluated using a Levene’s test which compared the variances of leg gap widths and determined whether the null hypothesis that strides varied similarly for an abnormal or irregular stride as for a normal stride could be rejected. Levene’s test checks for equality of variance among samples without requiring that the data be normally distributed [64].

The four individuals from whom data were collected were all students or instructors in an independent study course focused on gait monitoring. As a result, the authors’ home institution determined that this study was not human subjects research and did not require institutional review board (IRB) approval.

## 3. Results

The leg-based gait monitoring system was used to measure distance between the legs slightly above the knees (leg gap) and cadence as well as variability in these two gait parameters. The wearable system was tested on four different individuals and five different types of gait: normal, abnormal (fast, slow), and irregular (right antalgic, left antalgic).

### Leg Gap Width (as a Proxy for Stride Width)

The leg gap was measured in cm and calculated from the digital output obtained from the Arduino Uno during mid-stance (Figure 4). It serves as a proxy for stride width. Descriptive statistics for the leg gap calculated for five different stride types and four different individuals (subjects) based on the known characteristics of the Hall-effect sensor (Equation (3)) and ring magnet (Equation (2)) are summarized in Table 5.

Variability in leg gap was measured using four variability parameters: maximum (max), minimum (min), interquartile range (IQR), and standard deviation (SD) of leg gap values captured in mid-stance (Figure 3). Skewness and kurtosis of each group of data (subject, stride type) over multiple experiments were also measured to understand deviations from normality of distribution. Normality was tested using a Lilliefors test which is based on the Kolmogorov–Smirnov test for normality. The Lilliefors test assessed whether the null hypothesis that the sample data came from a normal parent population could be rejected at the 0.05 confidence level [65].

Nine of 20 groups failed the Lilliefors test. Therefore, subsequent statistical tests for the leg gap did not assume a normal distribution. Out of 20 groups of data, eight (40%) were highly skewed with absolute values greater than 1. Seven (35%) were moderately skewed with absolute values between 0.5 and 1, and the remaining five (25%) distributions were approximately symmetric with values ranging between −0.5 and 0.5 [66]. Only one distribution (corresponding to fast stride for Subject #1) was positively skewed. Nineteen distributions (95%) were negatively skewed, indicating that the bulk of the leg gaps were at the mean or larger than the mean. In a majority (89%) of these distributions, the mean was less than the median, a characteristic which is likely but not guaranteed of left skewed distributions. Kurtosis values indicated that a majority of the twenty sample distributions (80%) were more peaked than the normal distribution, indicating that outliers were not as likely among these data groups.

Because almost half of the leg gap distributions failed the Lilliefors test for normality, a non-parametric Kruskal–Wallis test was used to compare the medians of leg width associated with abnormal strides (fast, slow) and irregular strides (right antalgic, left antalgic) to normal strides for each individual. The Kruskal–Wallis test does not require that data be normally distributed in order to evaluate the null hypothesis that two or more sets of data come from the same parent population [63] and is equivalent to the Mann–Whitney test when only two groups are compared (as is the case here). Results of the Kruskal–Wallis test for the leg gap gait parameter are summarized in Table 6. In 10 of 16 (63%) pairwise comparisons between an abnormal or irregular stride and normal stride for a particular subject, the null hypothesis was rejected, meaning that the two strides were deemed significantly different. Comparing all five strides for each subject at once also revealed significant differences among the strides for each of the four subjects (Table 6).

To assess the role of outliers in detecting differences between each abnormal (fast, slow) or irregular (left antalgic, right antalgic) stride and the normal stride for each individual, these tests were repeated after outliers were removed. Outliers were removed using the standard statistical technique of eliminating all samples in each population which were less than 1.5-fold the IQR less than the first quartile or 1.5-fold the IQR greater than the third quartile [67]. After removing outliers from all groups (Table 7), 11 of 16 pairwise comparisons were statistically significant, thereby accurately predicting that 69% of abnormal or irregular strides were indeed different from normal strides.

To evaluate leg gap variability, Levene’s test was used to evaluate the null hypothesis that two groups (i.e., a normal stride and an abnormal or irregular stride for a particular subject) have equal variances. While the performance of Levene’s test is not as good as the Bartlett test which is also used to compare variance, Levene’s test is less sensitive to departures from normality and therefore is better matched to the leg gap data collected in this study [64]. Results from applying this test to the leg gap data are summarized in Table 8. Of the 16 tests comparing variance of abnormal or irregular stride to normal stride, 11 (69%) rejected the null hypothesis that the variances were equal, thereby accurately predicting the presence of strides that were not normal.

When outliers were removed from the data using the 1.5 IQR rule, the accuracy of predicting abnormal or irregular strides based on variability in leg gap increased from 69% to 81% (13 of 16). These results are summarized in Table 9.

A second gait parameter, cadence (steps per minute), was also evaluated using similar tests to those used to evaluate leg gap. Descriptive statistics for the cadence associated with five types of stride among four subjects are summarized in Table 10. Similar to leg gap, variability in cadence was measured using four variability parameters: maximum (max), minimum (min), interquartile range (IQR), and standard deviation (SD). Skewness and kurtosis were also calculated and normality tested using the Lilliefors test.

Five of 20 sample distributions failed the Lilliefors test and subsequent statistical tests for cadence and cadence variability did not assume a normal distribution. Out of 20 groups of data (i.e., five groups for each of four subjects), only 2 (10%) were highly skewed with an absolute value greater than 1. Five (25%) were moderately skewed with absolute values between 0.5 and 1, and the remaining 13 (65%) distributions were approximately symmetric with skewness values ranging between −0.5 and 0.5. Most data groups (12 of 20) were negatively skewed. Only five of the distributions indicated kurtosis (>4) that was much more peaked than a normal distribution.

Results of non-parametric Kruskal–Wallis tests used to compare the cadence of abnormal (fast, slow) or irregular (right antalgic, left antalgic) strides to normal strides are summarized in Table 11. In all (100%) of comparisons, the tests detected that strides that were not normal were significantly different than normal strides.

Unlike leg gap variability, variability in cadence was not particularly useful in differentiating strides (Table 12). Levene’s test rejected the null hypothesis that the variance of a stride that was not normal was equal to that of normal stride in only three of 16 cases (19%).

## 4. Discussion

The goal of this study was to evaluate the feasibility of using low-power, compact, high-resolution, analog (linear) Hall-effect sensors to differentiate normal gait from that which is abnormal in speed (fast, slow) or irregular in swing (left antalgic, right antalgic). To this end, two gait parameters (cadence and stride width) were evaluated using a self-contained, single-sensor system to answer two research questions.

### 4.1. Research Question 1 (RQ1)


*Does leg gap width captured using the wearable system make a sufficient proxy for stride width in terms of differentiating abnormal and irregular strides from normal?*


This question was evaluated based on the ability of the wearable sensor system to differentiate normal from abnormal or irregular strides. Leg gap (above the knee) is smaller than stride width and subject to smaller variations than stride width. However, it was anticipated that the high precision of the Hall-effect sensors (0.007 cm) in measuring leg gap width would compensate for the reduction in leg gap width and leg gap variability compared to corresponding stride width parameters (measured at the feet). When considering leg gap width alone, however, abnormal gait was significantly different from normal gait in only 69% of cases (Table 7) but when considering leg gap width variability alone (Table 9), 81% of abnormal stride types were significantly different from normal gait for a particular subject. When considering leg gap and its variability together, abnormal or irregular gait could be differentiated from normal gait in 94% of cases. Thus, leg gap is a promising proxy and alternative to measuring stride gap in terms of differentiating gait type. Despite the fact that none of the individuals who wore the device during this feasibility study were elderly, the usefulness of leg gap variability in differentiating abnormal and irregular gait from normal gait is consistent with studies of stride width during gait which show that both unusually small variability and unusually large variability can indicate balance deficits and other neurological disturbances [2,3]. In our data, two subjects compensated for simulated balance deficits in right antalgic and left antalgic strides (i.e., limps) via decreased leg gap variability and two compensated with increased leg gap variability. These differences are indicated in the boxplots of Figure 5 where the interquartile range (i.e., the height of the box) is higher in Subjects 1 and 2 for antalgic vs. normal strides whereas for Subjects 3 and 4, the interquartile range is smaller for antalgic strides.

### 4.2. Research Question 2 (RQ2)


*Can the wearable system capture cadence with sufficient accuracy to differentiate abnormal and irregular strides from normal strides?*


Previous research has indicated that cadence provides a necessary context for evaluating stride width variability. What constitutes extreme or concerning variability can change with cadence or gait speed of the individual [9]. Thus, an effective wearable gait monitoring system should capture cadence as part of measuring and assessing variability in stride width and related gait parameters. With a temporal resolution of 4 ms in resolving mid-stance events, the wearable Hall-effect sensor system demonstrated here was easily able to differentiate fast and slow strides from normal strides (Table 11) because they differed in cadence by 5 steps/minute or more. More subtle changes in cadence on the order of 1 step/minute brought on by limping (i.e., right antalgic and left antalgic gait) were also captured by the Hall-effect wearable. Thus, while using cadence to detect the difference among fast, slow, and normal strides was fairly trivial because of large differences in the mean cadence for these strides, more nuanced changes in cadence indicated by the similar means among normal, right antalgic, and left antalgic strides in Figure 6 can also be detected.

As with leg gap width, the precision of the Hall-effect wearable allowed for detecting small changes in both mean cadence times and nuances in the data distribution for different stride types. This raises the possibility that with high-resolution systems, cadence might also serve to support characterization of gait and detection of disturbances from normal gait. Given the evidence of how stable cadence is over an individual’s lifetime [10], however, it is not surprising that the variability in cadence across stride types for each individual offered no useful information in differentiating abnormalities or irregularities in stride (Table 12).

### 4.3. Benchmarking—Comparison to Existing Wearables

The feasibility of the leg-mounted Hall-effect wearable has been demonstrated for differentiating two types of abnormal gait (fast, slow) and two types of irregular gait (right antalgic, left antalgic) from normal gait. Such functionality is similar to other efforts (Table 2) which have successfully distinguished normal gait from that experienced by individuals recovering from knee surgery [37], suffering from Parkinson’s disease [19,39,49] or cerebrospinal meningitis [30], living with diabetes [48], or exercising simulated gait disturbances including obstacle courses [45] and toe joint restrictions [31]. Beyond this basic functionality, however, the Hall-effect approach offers some advantages and poses some disadvantages compared to existing systems.

*Pros:* The Hall-effect wearable offers high precision (on the order of tenths of millimeters) in detecting leg gap width compared to existing systems which offer precisions on the order of cm in detecting stride width [29,38]. While such precision is necessary to monitoring leg gap which is inherently less variable than stride width as measured at the feet, this increased precision nevertheless offers a greater potential to detect nuances in stride and stance variability. Further, this system offers the possibility of monitoring disturbances in gait with a single sensor at a single location on the body, whereas many existing systems (Table 1, Table 2 and Table 3) use multiple types of sensors or multiple sensors worn on several parts of the body to perform this function. The Hall-effect approach also offers some advantages in power consumption, lifetime, cost, and size. The Hall-effect sensor used in this work consumes appreciable power (on the order of 25 mW) because it requires a continuous supply current of approximately 10 mA to operate properly. However, Allegro Microsystems has replaced the 1302 with sensors from the 139X series [68] which offer a sleep mode that consumes only 0.08 mW, thereby offering significantly reduced overall power consumption in an application such as gait monitoring where the sensor need only be awake for a small fraction of time (i.e., when an individual is walking). Furthermore, unlike accelerometers, bend sensors, gyros, and similar sensors, the Hall-effect sensor has no moving parts, thus offering an increased lifetime among wearable gait sensors. Mounting along the thigh rather than the foot also decreases the mechanical stresses to which the sensor is subject, further increasing its lifetime. Finally, linear Hall-effect sensors are inexpensive (less than two dollars) compared to other sensors and smaller in size (Table 13). Since leg gap width can be estimated using a simple look-up table, the overhead (cost, power, size) involved in extracting useful information from a Hall-effect wearable is also much lower than most other sensors used in gait monitoring wearables.

*Cons:* Compared to the wealth of spatiotemporal information offered by accelerometers, gyros, and IMUs, the Hall-effect approach offers limited data. Its range is insufficient for monitoring stride length and other gait parameters that extend beyond 5 cm and temporal patterns of information are restricted to at or near mid-stance peaks. The leg wearable is also only suited to monitoring gait within an individual’s walking patterns and not across individuals. Comparing data between individuals is subject to too much variation in mounting location and other compliance related issues. These issues are minimized when sensors are integrated into the more rigid footprint provided by the shoe. However, one can easily imagine a highly inobtrusive Hall-effect sensor or sensors integrated into clothing which provide a low-overhead indicator of generalized gait disturbance and flag the potential need to monitor gait more comprehensively with the assistance of clinical or wearable systems that are designed to do so.

### 4.4. Future Work

The work presented here has focused on studying the feasibility of using Hall-effect sensors to identify gait disturbances. Statistical analyses suggest that leg gap, variability in leg gap, and cadence offer significant information for differentiating abnormal and irregular gait from normal gait. Future work should capitalize on the results of this work to explore efficient algorithms for recognizing gait disturbances based on cadence and leg gap data provided by these wearables. Time series analyses of the entire stride (rather than only the mid-stance peak) should also be considered in future efforts to explore the full functionality of the Hall-effect wearable in supporting everyday gait monitoring. Concurrently, we expect that future work will seek to miniaturize the wearable system such that it fully takes advantage of the small size and lower power sleep modes of the A139X series of Hall-effect sensors. Finally, the suitability of pairing a Hall-effect sensor wearable with a more comprehensive gait monitoring wearable will also be a topic of future research in this area.

## 5. Conclusions

The feasibility of using linear analog Hall-effect sensors for monitoring spatiotemporal gait parameters has been evaluated. Results show that leg gap and leg gap variability (as a proxy for stride width) and cadence data collected from these sensors can successfully detect disturbances in gait within individuals. These findings provide 69% accuracy in identifying gait disturbance when using leg gap alone, 81% accuracy when using leg gap variability alone, and 94% accuracy when using both in combination. For the controlled experiments used in this feasibility study, the temporal parameter of cadence offered 100% accuracy in detecting abnormal or irregular strides. Based on these initial findings, we can conclude that the Hall-effect sensor offers a promising low-cost, low-power alternative to classical accelerometer, gyro, or IMU-based wearables for flagging gait disturbance. These Hall-effect systems may also act as a supplement to classical systems for continuous monitoring of gait for the purpose of triggering more comprehensive monitoring by more complex wearables or by traditional gait analysis systems used in clinical settings. Future work will investigate the design and fabrication of more compact, integrated Hall-effect wearables, appropriate algorithms for recognizing and flagging abnormal and irregular gait, and potential integration with other types of sensors that offer information about other gait parameters including step length, foot orientation, and gait speed. A more compact, next generation sensor system should also be tested on a broader range of individuals to gauge generalizability of the usefulness of this device. Further, the capacity of the sensor system can also be expanded in future efforts to include multiple Hall-effect sensors mounted at different locations along the leg.

## Figures and Tables

**Figure 1 sensors-21-01206-f001:**
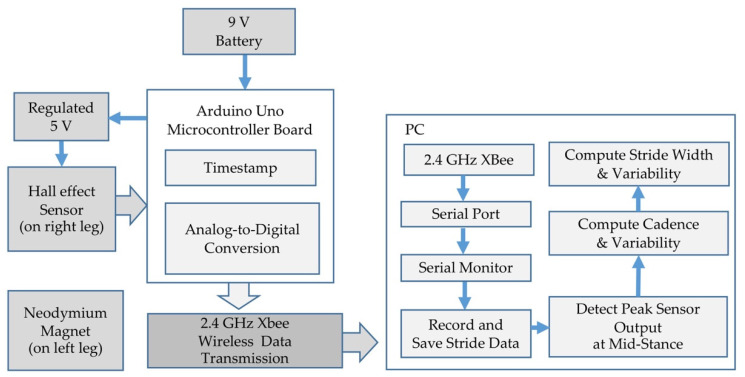
Block Diagram of Wearable Gait Monitoring System.

**Figure 2 sensors-21-01206-f002:**
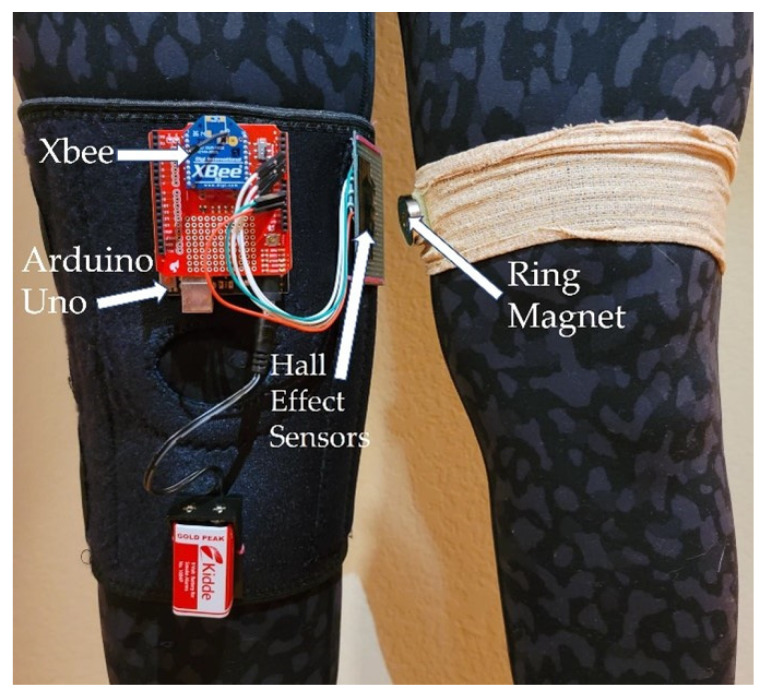
Prototype of Wearable Gait Monitoring System.

**Figure 3 sensors-21-01206-f003:**
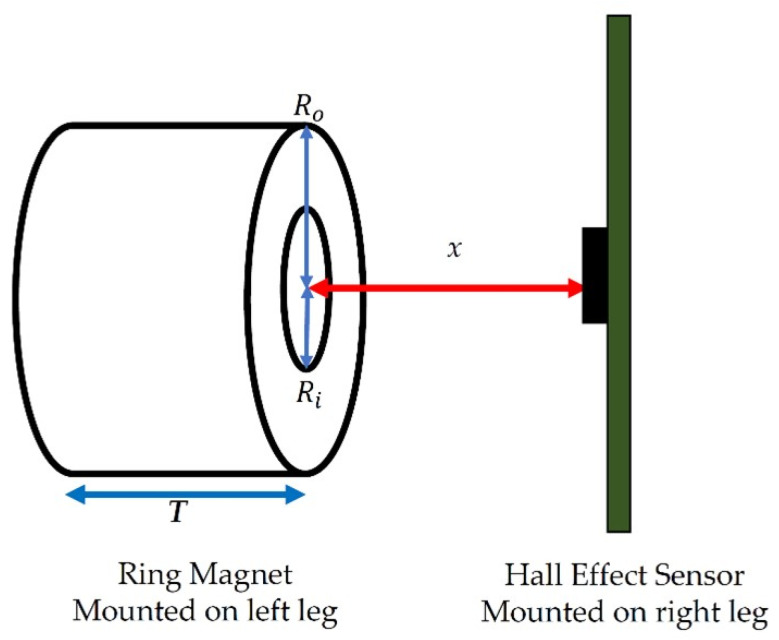
Sensor and Magnet Parameters.

**Figure 4 sensors-21-01206-f004:**
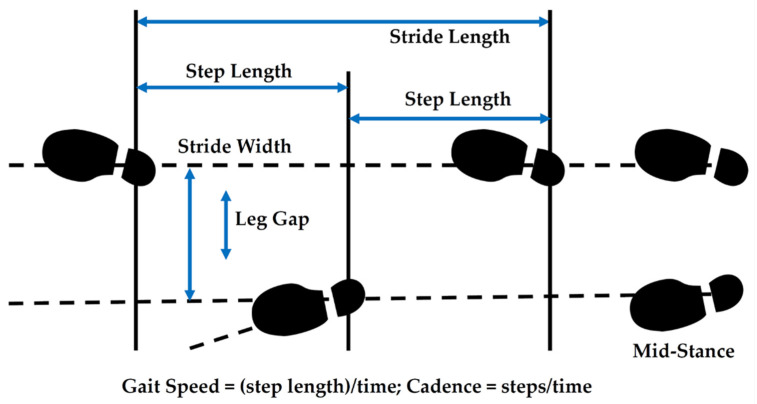
Gait Parameters.

**Figure 5 sensors-21-01206-f005:**
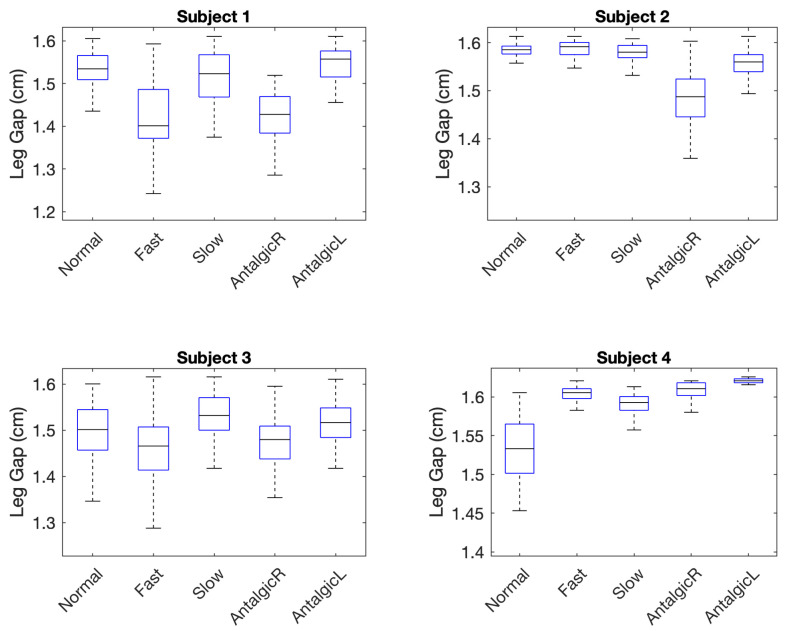
Boxplots indicating Leg Gap Variability for Four Subjects.

**Figure 6 sensors-21-01206-f006:**
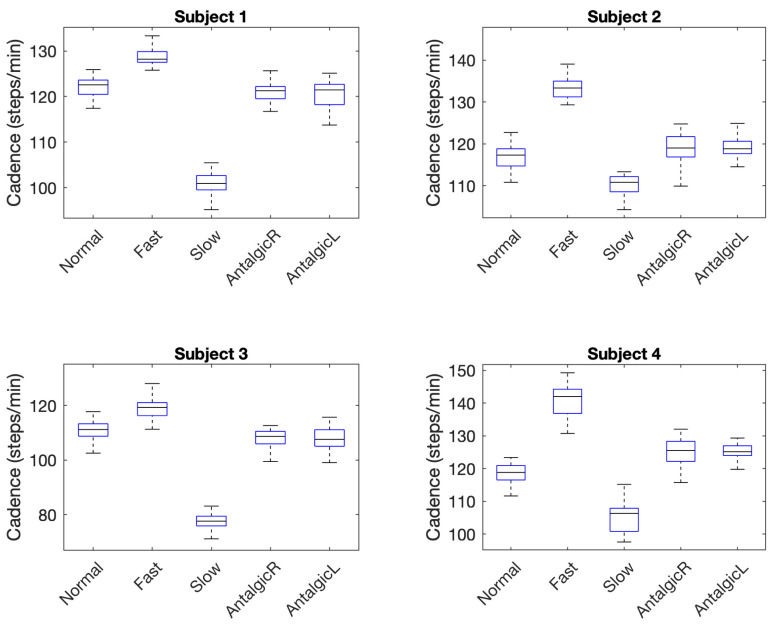
Boxplots of Cadence Indicating Differences in Stride Type.

**Table 1 sensors-21-01206-t001:** Wearable Sensors Used for Gait Analysis.

Year	Study	Sensors	Worn On	Function
2019	[29]	IMU	Ft	Measured stride width and stride length
2019	[30]	A	Ba	Measured toe-off and heel strike timing in the foot
2018	[31]	IR, IMU, F	Ft	Measured peak foot contact pressure, stance ratio, gait velocity
2017	[32]	F	Ft	Measured foot clearance for reducing falls among elderly
2017	[33]	IMG, EMG	L	Measured foot pressure distribution
2017	[34]	F	Ft	Evaluated gait asymmetry via foot contact pressure data
2015	[35]	IR	Ft	Measured foot position and orientation
2015	[36]	A	Ank	Measured multiple temporal gait parameters (e.g., cadence)
2015	[37]	F	Ft	Measured foot contact pressure during rehabilitation
2015	[38]	A, G, U	Ft	Measured step length and stride width
2014	[39]	IMU	S	Detected gait disturbances in Parkinson’s disease patients
2014	[40]	IMU	H, Ba, L	Detected onset of a turn
2014	[41]	IMU	Ft, S, Th	Detected gait phases
2013	[42]	IMU, IR	Ft	Measured step length and stride width
2013	[43]	G	S	Measured stride length and gait velocity
2013	[44]	A	W	Detected Parkinsonian step and measure step length
2010	[45]	F	Ft	Detected abnormal gait using ground contact forces
2009	[46]	G, A	Ft, L	Detected gait phases
2008	[19]	A, B, EF, F, G	Ft	Detected gait abnormalities among Parkinson’s disease patients
2007	[47]	F, IMU	Ft, L	Measured center of pressure, heel position, and ankle moment
2005	[48]	A, F	A, H, Ft, Sh, W	Detected gait abnormalities among diabetic patients
2004	[49]	G	Arm, Sh, Th	Detected gait abnormalities in Parkinson’s disease patients
2003	[50]	F	Ft	Measured sheer and vertical forces in the foot during gait
2002	[51]	G	Sh, Th	Measured stride length and gait velocity
2002	[32]	F	Ft	Measured contact pressures and foot contact time for diabetics
2002	[52]	U	Ft	Measured stance/swing duration and step/stride length
1997	[53]	F, G	Th	Measured stride length and walking velocity

A (accelerometers); B (bend sensors); EF (electric field sensors); F (force/pressure sensors); G (gyroscopes); IR (infrared range sensors); U (ultrasonic range sensors); IMU (inertial measurement units); EMG (electromyography); Ank (ankle); Arm; Ft (foot); Ba (back); H (head); L (legs); Th (thigh); S (shank); Sh (shoulders); W (waist).

**Table 2 sensors-21-01206-t002:** Wearable Sensors Used to Detect Abnormal Gait.

Study	Sensors	Worn	Approach
Bamberg et al. [19]	A, B, EF, F, G	Ft	Pitch, stride length, stride time, and percent stance time used to differentiate normal and Parkinsonian gait
Djuric-Jovicic et al. [39]	IMU	S	Rule-based data processing used to detect normal, short, and very short strides as well as freezing of gait in Parkinson’s disease patients
Li et al. [31]	IR, IMU, F	Ft	Peak pressure, stance ratio, stride length, walking velocity, and step-time variability used to compare toe-restrained vs. normal gait
Liu et al. [45]	F	Ft	Center of pressure and variation in foot contact forces used to detect fast, slow, and normal gait in obstacle and non-obstacle paths
Makino et al. [37]	F	Ft	Self-organizing maps (SOM) used to classify narrow vs. wide stride and slow vs. fast walking velocity for characterizing gait abnormality among patients who underwent total knee arthroplasty surgery
Petrofsky et al. [48]	A, F	Arm, H, Ft, Sh, W	Stance width and gait velocity found to be significantly different between normal and diabetic gait while walking a straight line and turning
Salarian et al. [49]	G	Arm, S, Th	Stride length and gait velocity; stance, double support, and gait cycle times used to detect abnormal gait among Parkinson’s disease patients
Yu et al. [30]	A	Ba	Kurtosis, crest factor, and mean stride interval used to distinguish abnormal and normal gait in cerebralspinal mengingitis patients

A (accelerometers); B (bend sensors); EF (electric field sensors); F (force/pressure sensors); G (gyroscopes); IR (infrared range sensors); IMU (inertial measurement units); Arm; Ft (foot); Ba (back); H (head); Th (thigh); S (shank); Sh (shoulders); W(waist).

**Table 3 sensors-21-01206-t003:** Performance of Wearable Sensors in Measuring Stride Width or Cadence/Gait Speed.

Study	Sensors	Worn	Performance
Cadence/Gait Speed/Gait Velocity			
Aminian et al. [51]	G	S, Th	0.006 m/sec accuracy relative to FSR reference
Lopez-Nava et al. [36]	A	Ank	4.6% accuracy in measuring cadence relative to GaitRite force/pressure sensing pad
Miyazaki et al. [53]	F, G	Th	±15% accuracy in measuring gait velocity
Salarian et al. [43]	G	S	−22.6 cm/s accuracy relative to a force-sensitive platform and a camera-based reference; 7.0 cm/s precision
Stride Width			
Hao et al. [29]	IMU	Ft	0.02 cm accuracy relative to a camera-based reference; 0.95 cm precision
Hung et al. [42]	IMU, IR	Ft	0.4 cm mean accuracy relative to marker reference
Liu et al. [45]	F	Ft	0.64 cm accuracy relative to a camera-based reference and force-sensitive plate reference; 0.1 cm precision
Weenk et al. [38]	A, G, U	Ft	1.2 cm accuracy relative to camera-based reference; 1.2 cm precision

A (accelerometers); F (force/pressure sensors); FSR (force sensitive resistor); G (gyroscopes); IR (infrared range sensors); U (ultrasonic range sensors); IMU (inertial measurement units—include gyroscopes and accelerometers); Ank (ankle); Ft (foot); Th (thigh); S (shank).

**Table 4 sensors-21-01206-t004:** Gait Types Used to Test the Prototype Gait Monitoring System.

Gait Label	Type	Description
Normal	Normal	Walk at a normal pace, one that requires no conscious attention or focus on the nature or speed of gait.
Fast	Abnormal	Walk at a brisk pace, as if in a hurry.
Slow	Abnormal	Walk slowly, as if enjoying a sunny day.
Right Antalgic	Irregular	Walk in such a way that stance/swing ratio on the right side of the body is smaller than normal gait—by walking with the left shoe off. Simulates a right limp.
Left Antalgic	Irregular	Walk in such a way that stance/swing ratio on the left side of the body is smaller than normal gait—by walking with the right shoe off. Simulates a left limp.

**Table 5 sensors-21-01206-t005:** Descriptive Statistics and Normality of Leg Gap Data.

Type	*N*	Leg Gap Width (cm)					Skew	Kurtosis	Normal *
		Mean	SD	IQR	Max	Min			
Subject #1									
Normal	37	1.53	0.046	0.057	1.61	1.44	−0.24	2.33	Y
Fast	38	1.42	0.090	0.114	1.59	1.24	0.22	2.54	Y
Slow	48	1.50	0.091	0.099	1.61	1.12	−1.60	5.61	N
AntalgicR	37	1.41	0.089	0.086	1.52	1.46	−1.09	3.43	N
AntalgicL	40	1.55	0.040	0.061	1.29	1.14	−0.67	2.55	N
Subject #2									
Normal	44	1.58	0.017	0.017	1.61	1.50	−2.50	13.50	N
Fast	38	1.59	0.023	0.025	1.61	1.52	−1.12	3.91	N
Slow	44	1.58	0.016	0.025	1.61	1.53	−0.43	4.40	Y
AntalgicR	42	1.48	0.065	0.079	1.60	1.25	−1.01	5.64	Y
AntalgicL	46	1.55	0.048	0.036	1.61	1.37	−1.60	5.86	N
Subject #3									
Normal	51	1.49	0.074	0.088	1.60	1.28	−0.68	3.08	Y
Fast	41	1.47	0.072	0.093	1.62	1.29	−0.25	2.91	Y
Slow	53	1.53	0.054	0.071	1.62	1.36	−0.83	3.68	Y
AntalgicR	50	1.48	0.063	0.071	1.60	1.25	−0.84	5.13	Y
AntalgicL	49	1.51	0.050	0.064	1.61	1.38	−0.55	3.18	Y
Subject #4									
Normal	38	1.53	0.044	0.064	1.61	1.41	−0.46	3.26	Y
Fast	32	1.60	0.012	0.013	1.62	1.57	−1.01	3.45	N
Slow	34	1.59	0.016	0.018	1.61	1.55	−0.81	3.33	Y
AntalgicR	36	1.61	0.011	0.017	1.62	1.58	−0.89	3.07	N
AntalgicL	41	1.62	0.005	0.005	1.63	1.60	−3.11	17.10	N

* Evaluated using the Lilliefors (based on Kolmogorov–Smirnov test) normality test.

**Table 6 sensors-21-01206-t006:** Detection of Abnormal or Irregular Strides Using Leg Gap (Kruskal–Wallis test).

Fast Strides		Slow Strides		Right Antalgic		Left Antalgic		All Strides	
Subject #1									
*H(1)* = 27.53 *p* = 0.000	***	*H(1)* = 1.93 *p* = 0.165		*H(1)* = 40.08 *p* = 0.000	***	*H(1)* = 0.34 *p* = 0.557		*H(4)* = 71.46 *p* = 0.000	***
Subject #2									
*H(1)* = 1.71 *p* = 0.190		*H(1)* = 1.42 *p* = 0.233		*H(1)* = 51.48 *p* = 0.000	***	*H(1)* = 26.49 *p* = 0.000	***	*H(4)* = 100.52 *p* = 0.000	***
Subject #3									
*H(1)* = 4.19 *p* = 0.041	*	*H(1)*= 7.02 *p* = 0.008	**	*H(1)* = 2.83 *p* = 0.093		*H(1)* = 1.19 *p* = 0.276		*H(4)* = 33.32 *p* = 0.000	***
Subject #4									
*H(1)* = 44.59 *p* = 0.000	***	*H(1)* = 34.44 *p* = 0.000	***	*H(1)* = 50.03 *p* = 0.000	***	*H(1)* = 58.99 *p* = 0.000	***	*H(4)* = 135.58 *p* = 0.000	***

* *p* < 0.05; ** *p* < 0.01; *** *p* < 0.001.

**Table 7 sensors-21-01206-t007:** Detection of Abnormal or Irregular Strides Using Leg Gap *(Kruskal–Wallis test, with outliers removed).*

Fast Strides		Slow Strides		Right Antalgic		Left Antalgic		All Strides	
Subject #1									
*H*(1) = 10.70 *p* = 0.001	**	*H*(1) = 2.62 *p* = 0.105		*H*(1) = 16.12 *p* = 0.000	***	*H(1)* = 9.13 *p* = 0.003	**	*H(4)* = 55.45 *p* = 0.000	***
Subject #2									
*H*(1) = 3.22 *p* = 0.073		*H*(1) = 1.52 *p* = 0.217		*H*(1) = 50.93 *p* = 0.000	***	*H(1)* = 23.10 *p* = 0.000	***	*H(4)* = 101.95 *p* = 0.000	***
Subject #3									
*H*(1) = 5.03 *p* = 0.025	*	*H*(1) = 7.35 *p* = 0.007	**	*H*(1) = 2.98 *p* = 0.084		*H(1)* = 1.63 *p* = 0.202		*H* (4) = 36.92 *p* = 0.000	***
Subject #4									
*H*(1) = 44.83 *p* = 0.000	***	*H(1)* = 34.64 *p* = 0.000	***	*H(1)* = 49.97 *p* = 0.000	***	*H(1)* = 59.23 *p* = 0.000	***	*H(4)* = 138.14 *p* = 0.000	***

* *p* < 0.05; ** *p* < 0.01; *** *p* < 0.001.

**Table 8 sensors-21-01206-t008:** Detection of Abnormal or Irregular Strides using Leg Gap Variability *(Levene’s test)*.

Subject	Fast Strides		Slow Strides		Right Antalgic		Left Antalgic	
1	*F* = 14.09 *p* = 0.0003	***	*F* = 7.28 *p* = 0.009	**	*F* = 8.04 *p* = 0.006	**	*F* = 1.19 *p* = 0.279	
2	*F* = 4.16 *p* = 0.048	*	*F* = 0.34 *p* = 0.562		*F* = 29.89 *p* = 0.000	***	*F* = 16.79 *p* = 0.0001	***
3	*F* = 0.04 *p* = 0.844		*F* = 3.92 *p* = 0.051		*F* = 1.56 *p* = 0.214		*F* = 5.01 *p* = 0.028	*
4	*F* = 29.31 *p* = 0.000	***	*F* = 22.56 *p* = 0.000	***	*F* = 33.73 *p* = 0.000	***	*F* = 59.24 *p* = 0.000	***

* *p* < 0.05; ** *p* < 0.01; *** *p* < 0.001.

**Table 9 sensors-21-01206-t009:** Detection of Abnormal or Irregular Strides Using Leg Gap Variability (Levene’s test, with outliers removed).

Subject	Fast Strides		Slow Strides		Right Antalgic		Left Antalgic	
1	*F* = 14.37 *p* = 0.0004	***	*F* = 7.42 *p* = 0.008	**	*F* = 8.82 *p* = 0.005	**	*F* = 2.09 *p* = 0..156	
2	*F* = 6.80 *p* = 0.011	*	*F* = 4.15 *p* = 0.045	*	*F* = 49.57 *p* = 0.000	***	*F* = 16.57 *p* = 0.000	***
3	*F* = 0.03 *p* = 0.87		*F* = 4.67 *p* = 0.033	*	*F* = 2.57 *p* = 0.112		*F* = 7.28 *p* = 0.008	**
4	*F* = 35.84 *p* = 0.000	***	*F* = 27.19 *p* = 0.000	***	*F* = 38.99 *p* = 0.000	***	*F* = 78.45 *p* = 0.000	***

* *p* < 0.05; ** *p* < 0.01; *** *p* < 0.001.

**Table 10 sensors-21-01206-t010:** Descriptive Statistics and Normality of Cadence Data.

Type	Samples	Cadence [Steps/Min]					Skewness	Kurtosis	Normal *
		Mean	SD	IQR	Max	Min			
Subject #1									
Normal	29	122.22	2.06	3.13	125.92	117.42	−0.37	2.73	Y
Fast	27	127.97	3.45	2.35	133.33	116.39	−2.07	7.73	N
Slow	33	100.74	2.54	3.15	105.45	95.16	−0.49	2.78	Y
AntalgicR	30	121.15	2.32	2.68	125.66	116.73	0.25	2.47	Y
AntalgicL	34	120.50	3.03	4.47	125.13	113.74	−0.59	2.39	N
Subject #2									
Normal	41	117.15	3.64	4.12	127.52	108.11	0.39	4.36	Y
Fast	32	134.17	4.34	3.77	145.81	129.31	1.49	4.72	N
Slow	43	110.33	2.97	3.63	121.09	104.26	0.50	5.61	N
AntalgicR	36	118.75	4.06	4.86	124.74	108.50	−0.82	3.29	Y
AntalgicL	42	119.34	2.85	2.96	126.58	112.36	0.29	3.54	Y
Subject #3									
Normal	43	110.75	3.73	4.55	117.76	101.35	−0.67	3.24	N
Fast	35	119.41	4.39	4.78	132.01	111.32	0.66	3.79	Y
Slow	48	77.44	3.30	3.57	86.15	70.01	−0.24	3.33	Y
AntalgicR	44	108.44	4.14	4.59	120.48	99.50	0.48	4.52	Y
AntalgicL	43	107.77	4.01	4.01	115.72	99.09	−0.14	2.36	Y
Subject #4									
Normal	32	118.40	2.93	4.40	123.33	111.63	−0.63	2.98	Y
Fast	26	141.06	4.72	7.40	149.25	130.72	−0.24	2.43	Y
Slow	28	105.53	4.95	7.07	115.16	97.56	0.10	2.27	Y
AntalgicR	29	125.10	4.00	6.14	132.01	115.72	−0.43	2.78	Y
AntalgicL	35	125.26	2.53	2.98	129.31	119.28	−0.46	2.88	Y

* Evaluated using the Lilliefors (based on Kolmogorov–Smirnov test) normality test.

**Table 11 sensors-21-01206-t011:** Detection of Abnormal or Irregular Strides using Cadence [steps/minute] (Kruskal–Wallis test).

Subject	Fast Strides		Slow Strides		Right Antalgic		Left Antalgic	
1	*H* (1) = 30.10 *p* = 0.000	***	*H* (1) = 45.60 *p* = 0.000	***	*H* (1) = 3.89 *p* = 0.049	*	*H* (1) = 5.03 *p* = 0.025	*
2	*H* (1) = 53.20 *p* = 0.000	***	*H* (1) = 47.95 *p* = 0.000	***	*H* (1) = 5.16 *p* = 0.023	*	*H* (1) = 10.11 *p* = 0.000	***
3	*H* (1) = 46.62 *p* = 0.000	***	*H* (1) = 67.32 *p* = 0.000	***	*H* (1) = 10.86 *p* = 0.001	**	*H* (1) = 11.09 *p* = 0.0009	***
4	*H* (1) = 42.32 *p* = 0.000	***	*H* (1) = 41.57 *p* = 0.000	***	*H* (1) = 31.17 *p* = 0.000	***	*H* (1) = 43.36 *p* = 0.000	***

Cadence of Stride is significantly different from normal (* *p* < 0.05; ** *p* < 0.01; *** *p* < 0.001).

**Table 12 sensors-21-01206-t012:** Detection of Abnormal or Irregular Strides Using Cadence Variability (Levene’s test).

Subject	Fast Strides		Slow Strides		Right Antalgic		Left Antalgic	
1	*F* = 0.53 *p* = 0.487		*F* = 0.90 *p* = 0.347		*F* = 0.47 *p* = 0.496		*F* = 5.48 *p* = 0.023	*
2	*F* = 0.36 *p* = 0.551		*F* = 0.48 *p* = 0.402		*F* = 1.04 *p* = 0.312		*F* = 0.92 *p* = 0.341	
3	*F* = 0.41 *p* = 0.524		*F* = 9.35 *p* = 0.493		*F* = 0.09 *p* = 0.763		*F* = 0.89 *p* = 0.345	
4	*F* = 7.57 *p* = 0.008	**	*F* = 6.69 *p* = 0.0013	**	*F* = 3.33 *p* = 0.073		*F* = 0.65 *p* = 0.424	

* *p* < 0.05; ** *p* < 0.01; *** *p* < 0.001.

**Table 13 sensors-21-01206-t013:** Typical Constraints of Sensors Used in Wearable Gait Monitors.

Parameter (Sensor Type)	Example	Power [mW] ^1^	Cost [USD] ^4^	Size [cm] ^5^	Weight [g]
Bending/Vibration (PVDF Strips)	TE Connectivity LDT0 [69] used in [19]	NA	$2.69	2.5 × 1.3	NA
Angular rate, orientation (gyroscope)	Analog Devices ADXRS290 similar to ADXRS150 [70] used in [19]	39 (0.4) ^2^	$16.06	0.6 × 0.4	NA
Acceleration, angular rate, orientation (IMU)	XSens MTi-1 [71]	45	$139	1.2 × 1.2	0.66
Distance (Hall effect)	Allegro 139X [68] replacement for sensors used here	9.6 (0.08) ^2^	$1.20	0.8 × 0.3	0.014
Distance (Infrared IR)	Sharp GP2YOA41SKOF [72] used in [61]	60	$9.93	4.4 × 1.9	3.5
Distance (Ultrasonic)	URM07 [73]	25 (0.07) ^2^	$5.59	2.7 × 2.7	4.2
Force/Pressure (Force-Sensitive Resistor)	FSR-400 [74] used in [19]	2.5 ^3^	$4.40	4.0 × 4.0	NA
Linear Acceleration (accelerometer)	Analog Devices ADXL202E [75] used in [19]	3.0	$19.72	0.7 × 0.7	5
Object Presence (electric field sensors)	Motorola MC33794DH [71] used in [19]	98	$24 (est)	1.6 × 1.1	NA

^1^ Power consumption provided at typical supply voltages (as specified in datasheet). ^2^ In measurement (sleep) or measurement (non-actuated) mode at typical supply voltage. ^3^ Based on 10 k measurement resistance when bent and a basic measurement circuit. ^4^ Cost estimated at quantities of 100 units. ^5^ Thickness was either not specified or is small compared to remaining two dimensions; NA: not available.

## Data Availability

The data presented in this study will be made openly available upon acceptance and publication in Open Science Framework at (DOI 10.17605/OSF.IO/R4TF9).

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
