# Peer review of "Abnormal Gait Detection Using Wearable Hall-Effect Sensors"

_sensors, 2021, doi:10.3390/s21041206_

Round 1

Reviewer 1 Report

This is a significant paper on abnormal gait analysis with a wearable system. The context on gait monitoring and analysis as well as wearable systems has been extensively discussed. The low-power Hall effect sensors and a compact magnet have been used to analyze the gait parameters. The experimental results have been presented and deeply analyzed. The current version can be further improved by considering the following remarks:

1) In I troduction, the advantages of the proposed wearable system using Hall effect sensors related to the existing work on gait analysis (e.g. classical systems with accelerometers, pressure analysis, other studies on Hall effect sensors) should be more precisely indicated.   

2) The raw data measured from wearable systems are usually noisy. The data preprocessing process for noise filtration can be more discussed.

3) For data analysis (e.g. leg gap width and cadence estimations), the proposed procedure seems too simplified with statistical analysis. For different subjects, the results could be different. If a learning database on gait analysis can be applied to the procedure, the final results could be more relevant. Also, time series analysis can also be helpful for providing relevant results.

4) For analysis of results, it could be interesting to make a comparison between the proposed system and tohse obtained from the existing methods with other sensors.

Reviewer 2 Report

Please find the reviewer's comment as an attachment.

Reviewer 3 Report

In order to improve the scientific soundness of the paper, the following updates must be added:

  • the proposed setup must be correlated with existing solutions: explain advantages and disadvantages of the proposed setup regarding the existing ones
  • regarding the evaluation: only 4 users for testing are enough to validate the solution? How different people (age, sex, medical conditions) were simulated through the 4 users used for tests?
  • obtained results must be correlated with the existing ones
  • section 2 starts with a figure. It must start with an introduction / explication; also, figures must be placed after their first appearance in the text.

Round 2

Reviewer 1 Report

The remarks of the reviewers have been taaken into account in the new version. The paper can be accepted for publication.

Author Response

Thank you very much for your time in reviewing this paper. 

Reviewer 2 Report

Please find the reviewer's comment as the attachment.

Author Response

Thank you very much for your time in reviewing this paper. We have made the revisions you requested. Thank you for your patience. 

Reviewer 3 Report

The authors added detailed responses for my comments. Thus, I recommend to publish the paper.

Author Response

(The authors gave the same response as above.)
